# A Regulatory Axis between Epithelial Splicing Regulatory Proteins and Estrogen Receptor α Modulates the Alternative Transcriptome of Luminal Breast Cancer

**DOI:** 10.3390/ijms23147835

**Published:** 2022-07-16

**Authors:** Jamal Elhasnaoui, Giulio Ferrero, Valentina Miano, Lorenzo Franchitti, Isabella Tarulli, Lucia Coscujuela Tarrero, Santina Cutrupi, Michele De Bortoli

**Affiliations:** 1Department of Clinical and Biological Sciences, School of Medicine, University of Turin, Orbassano, 10043 Turin, Italy; jamal.elhasnaoui@unito.it (J.E.); giulio.ferrero@unito.it (G.F.); lorenzo.franchit@unito.it (L.F.); isabella.tarulli@edu.unito.it (I.T.); santina.cutrupi@unito.it (S.C.); 2Candiolo Cancer Institute-FPO, IRCCS, Candiolo, 10060 Turin, Italy; 3Department of Computer Science, School of Life Sciences, University of Turin, 10149 Turin, Italy; 4Division of Cellular and Molecular Pathology, Department of Pathology, University of Cambridge, Addenbrooke’s Hospital, Cambridge CB2 0QQ, UK; vm403@cam.ac.uk; 5Center of Genomic Science of IIT@SEMM, Italian Institute of Technologies, 20139 Milan, Italy; lucia.coscujuelatarrero@unito.it

**Keywords:** breast cancer, ESRP1, ESRP2, alternative splicing, EMT splicing signature, Rac1, Rac1b

## Abstract

Epithelial splicing regulatory proteins 1 and 2 (ESRP1/2) control the splicing pattern during epithelial to mesenchymal transition (EMT) in a physiological context and in cancer, including breast cancer (BC). Here, we report that *ESRP1*, but not *ESRP2*, is overexpressed in luminal BCs of patients with poor prognosis and correlates with estrogen receptor α (ERα) levels. Analysis of ERα genome-binding profiles in cell lines and primary breast tumors showed its binding in the proximity of *ESRP1* and *ESRP2* genes, whose expression is strongly decreased by ERα silencing in hormone-deprived conditions. The combined knock-down of ESRP1/2 in MCF-7 cells followed by RNA-Seq, revealed the dysregulation of 754 genes, with a widespread alteration of alternative splicing events (ASEs) of genes involved in cell signaling, metabolism, cell growth, and EMT. Functional network analysis of ASEs correlated with *ESRP1*/*2* expression in ERα+ BCs showed RAC1 as the hub node in the protein–protein interactions altered by ESRP1/2 silencing. The comparison of ERα- and ESRP-modulated ASEs revealed 63 commonly regulated events, including 27 detected in primary BCs and endocrine-resistant cell lines. Our data support a functional implication of the ERα-ESRP1/2 axis in the onset and progression of BC by controlling the splicing patterns of related genes.

## 1. Introduction

The comparison of tumor tissues to their healthy counterparts revealed the presence of aberrant Alternative Splicing (AS) patterns, a novel cancer hallmark [1,2,3]. Indeed, AS analyzed in >8000 tumors from The Cancer Genome Atlas (TCGA) revealed thousands of cancer-specific AS variants [4,5], whereas other isoform-based studies associated the relative abundance of AS isoforms with tumor stage and patient survival [6].

Breast cancer (BC) is a heterogeneous disease with distinct subtypes associated with different phenotypes, response to therapy, and disease outcomes [7]. These subtypes are classified based on the expression of estrogen receptor-α (ERα), progesterone receptor (PR), and human growth-factor receptor-2 (HER2) [8]. The luminal ERα+ subtype represents two-thirds of BC cases, followed by HER2-amplified BC, and by the Basal-like triple-negative subtype [7].

AS was identified as a pivotal process involved in BC onset and progression [9]. In this context, AS was related to several cellular processes, including proliferation, angiogenesis, apoptosis, adapting metabolism, cancer cell invasion, metastasization, and resistance to hormonal therapy [9,10,11].

AS dysregulation in cancer has been linked to dysfunctional expression or copy number alteration of RNA-binding proteins (RBPs) or mutations of their target RNAs [12]. For example, the RBPs, epithelial-splicing regulatory proteins (ESRP) 1 (ESRP1) and 2 (ESRP2), represent the core regulators of AS in epithelial cells. These proteins maintain the epithelial cell phenotype and prevent the epithelial to mesenchymal transition (EMT) process [13]. Conversely, *ESRP1* and *ESRP2* overexpression in mesenchymal cells induces isoform switching events, promoting an EMT-reverse process called mesenchymal to epithelial transition [14].

Studies have revealed the clinical significance of ESRP1 and ESRP2 in tumor progression and metastasis [15]. Other studies revealed that the expression of some ESRPs-regulated exons correlates with a favorable prognosis, whereas the expression of ESRP2 is not associated with the overall survival of cancer patients [16]. In contrast, ESRP1 is a marker of poor prognosis in ERα+ but not ERα− BC, and its depletion in ERα+ BC models reduces tumor growth [17].

Our group demonstrated that ERα chromatin interactions can occur independently of estrogenic stimuli (apoERα), and these interactions functionally regulate genes involved in epithelial cell growth and development [18]. We further demonstrated that these interactions occur in endocrine-resistant cells and regulate the expression of genes relevant to BC progression [19,20]. More recently, we showed that a high proportion of apoERα-regulated genes and isoforms are composed of RBPs, most of which (75%) were lost upon ERα depletion, including ESRP1 and ESRP2 [21].

In this work, we investigate transcriptional and AS alteration effects of the combined ESRP1 and ESRP2 silencing in the ERα+ model, MCF-7 BC cells. The analysis confirmed the activity of these proteins in regulation of EMT genes, as well as genes involved in the inflammatory response. Integration with ERα ChIP-Seq and RNA-Seq experiments of estrogen-deprived MCF-7 showed that ERα binds in the proximity of *ESRP1* and *ESRP2* genes and regulates their expression in the absence of ligands. A subset of ESRP1/2-regulated ASEs was also identified in this experimental condition and confirmed in primary BC, reinforcing the hypothesis of a functional ERα-ESRP1/2 regulatory axis involved in BC tumorigenesis.

## 2. Results

### 2.1. ESRP1 and ESRP2 Expression Is Altered in ERα+ BC and It Is Regulated by ERα

To investigate the relationship between ERα, *ESRP1,* and *ESRP2* expressions in BC, *ESRP1* and *ESRP2* copy number alteration status and mRNA levels were analyzed in 774 ERα+ BC samples from TCGA (Appendix A). The genomic locus harboring the *ESRP1* gene was characterized by a copy number gain (60% of samples), while the *ESRP2* gene locus was characterized by a significant heterozygous deletion (62% of samples) (Appendix A). The analysis of *ESRP1* and *ESPR2* mRNA levels revealed their relationship with specific molecular and clinical features of tumor samples (Figure 1a and Appendix A). Specifically, the expression levels of both genes were correlated with the fraction of genome altered (*p* < 0.0001), particularly for *ESRP1* (rho = 0.50, *p* < 0.0001) (Figure 1b), and were higher in luminal B tumors compared to the other BC subtypes (*p* < 0.0001) (Appendix A). The expression of both genes was significantly lower in patients presenting micrometastasis (*p* < 0.01 and *p* < 0.05 for *ESRP1* and *ESRP2*, respectively). In contrast, only *ESRP1* expression was related to the menopause status (*p* < 0.05) and overall survival (OS) status (*p* < 0.05) (Appendix A), as previously observed [17].

Furthermore, from this analysis, we observed that both ESRP1 and ESRP2 genes confirmed a clear correlation with the mRNA level of *ESR1* gene (*p* < 0.001) (Figure 1c), as confirmed by correlation analysis (rho = 0.30 and 0.20 for ESRP1 and ESRP2, respectively; *p* < 0.0001) (Appendix A). In addition, the analysis of public ERα ChIP-Seq experiments in BC cell lines and primary tissues confirmed ERα binding in the proximity of both genes (Figure 1d). Regarding *ESRP2* gene, promotorial and an intronic ERα binding sites were observed in estrogens-treated or hormone-deprived MCF-7 cells [22], in tamoxifen-responsive and resistant cell lines (MCF-7, BT474), and primary BCs [23]. ERα binding at *ESRP1* gene was observed at gene promoter, but only in a subset of drug-responsive MCF-7 and primary BC tissues. Indeed, by showing the significant downregulation of *ESRP1* and *ESRP2* gene expression upon ERα silencing in hormone-deprived MCF-7 cells [19], we confirmed that their expression depends on ERα signaling.

### 2.2. Effects of ESRP1 and ESRP2 Knock-Down in MCF-7 Cells

To characterize the ESRP1/2 regulatory activity in ERα+ BC cells, we analyzed the transcriptomic effects of the combined ESRP1/2 silencing (siESRP1/2) in MCF-7 cells (Appendix A) by RNA-seq. Differential expression analysis with respect to the control condition revealed 754 differentially expressed (DE) genes, of which 422 are downregulated and 332 are upregulated (Figure 2a and Appendix A). Downregulated genes are functionally enriched in terms related to *type I interferon signaling pathways*, *regulation of cell adhesion*, *morphogenesis of an epithelium*, and *regulation of cell proliferation*. Conversely, upregulated genes are enriched in processes such as *regulation of exocytosis*, *regulation of vesicle mediated transport*, and *extracellular matrix organization* (Figure 2b,c and Appendix A).

To decipher the role of ESRP1 and ESRP2 in controlling AS in MCF-7 cells, a differential AS analysis was performed using rMATS [24,25]. The analysis revealed that ESRP1/2 knock-down induces AS changes in 788 genes with 1052 significantly regulated (*p* < 0.05) AS events (ASEs): 810 (77%) exon skipping (ES), 109 (10%) mutually exclusive exons (MXE), 62 (6%) alternative 5′ splice sites (A5SS), 48 (5%) alternative 3′ splice sites (A3′SS), and 23 (2%) intron retention (IR) events (Figure 2c,d and Appendix A). The top 50 significant ASEs included 45 ES and 5 MXE events (Figure 2e and Appendix A). Functional analysis of ASE-harboring genes showed an enrichment of *microtubule cytoskeleton organization*, *cell projection assembly*, and *regulation of cell-cycle processes* (Figure 2f and Appendix A). These processes were mainly characterized by ES events, whereas genes with MXE events were enriched in metabolic processes, including the *carbohydrate derivative biosynthetic process*, *the cholesterol biosynthetic process*, and the *EMT process* (Figure 2f and Appendix A). Accordingly, our analysis confirmed previously published EMT-related ASEs involving known ESRP-target genes [17,26], including *USO1* (ES), *SLK* (ES), *ARGFAP2* (ES), *FLNB* (ES and A5′SS), *SCRIB* (ES), *MAGI1* (ES), *RALGPS2* (ES), *ENAH* (ES), *ARHGEF10L* (ES), and *RAC1* (ES) (Appendix A).

Moreover, to determine whether ESRPs are directly involved in the control of the AS changes induced by their combined silencing, an RBP-binding motif enrichment analysis was performed for 105 splicing factors (SFs) (122 binding motifs), including ESRP1 and ESRP2. As expected, RBP-binding motif analysis showed an over-representation of ESRP-binding motifs for 694 ASEs, including 540 ES events (51.33%) and 70 MXE events (6.65%) (Appendix A). In ES events, ESRP binding motif showed exonic enrichment, whereas, in MXE events, it was also enriched in the upstream intronic sequences (Appendix A).

To explore the role of ESRP-modulated ASEs in primary BC specimens, the inclusion levels (PSIs) of exons involved in the identified ASEs were analyzed in public data from SpliceSeq [27], revealing 327 ASEs (264 ESs, 50 MXEs, 7 A5SSs, and 6 A3SSs) detectable in tumor samples (Appendix A). Among them, the inclusion levels of exons involved in 122 ASEs (103 ES, 15 MXE, 2 A5SS, and 2 A3SS) were significantly correlated (*p* < 0.05) with the mRNA levels of both *ESRP1* and *ESRP2* genes in BCs (Appendix A). Furthermore, 40 ASEs were differentially expressed relative to the median mRNA expression of *ESRP1* (ESRP1 high versus ESRP1 low) and *ESRP2* genes (ESRP2 high versus ESRP2 low) (Appendix A). In addition, the correlation between the PSIs of each ESRPs-modulated ASE and overall and disease-free survival times was explored, revealing 36 ASEs significantly associated with overall survival (OS) or disease-free survival (DFS) times of ERα+ BC subjects (Appendix A). In particular, higher inclusion of 16 and 4 ASEs is associated with longer and shorter OS time, respectively. Similarly, higher inclusion of 8 and 8 ASEs is associated with longer and shorter DFS time, respectively. The top significantly associated event with patient OS is the inclusion of exon 2 of the *SPINT2* gene (HR = 0.73; log-rank *p*-value = 5.73 × 10^−4^), while the top significantly associated event with DFS is the skipping of exon 4 of the *LSR* gene (HD = 0.96; log-rank *p*-value = 2.22 × 10^−4^) (Appendix A).

### 2.3. Validation of ASEs Regulated by ESRP1 and ESRP2

Among the identified ASEs, the top four significant ES events from rMATS analysis (Figure 3a) were selected for experimental validation via qualitative PCR. The validation was performed using couples of primers directed against the flanking exons of the alternative exons involved in the ASEs (Table 1, see Materials and Methods section). Interestingly, in addition to the significant correlation with *ESRP1* and *ESRP2* mRNA levels in ER+ BCs, *USO1* (r = 0.45, *p* = 6.12 × 10^−38^; r = 0.45, *p* = 7.12 × 10^−39^), *MYOF* (r = −0.30, *p* = 3.02 × 10^−16^; r = −0.23, *p* = 2.19 × 10^−10^), *SCRIB* (r = −0.34, *p* = 8.63 × 10^−22^; r = −0.10, *p* = 0.03), and *RAC1* (r = 0.10, *p* = 0.004; r = 0.15, *p* = 0.0001), for *ESRP1* and *ESRP2*, respectively, these ASEs also exhibit a significant differential inclusion relative to the median mRNA expression of *ESRP1* and *ESRP2* in these tumors (Figure 3b and Appendix A). The PCR results confirmed that ESRP1/2 combined silencing induces higher inclusion levels of exons 17 in *MYOF* and 16 in *SCRIB*, lower inclusion levels of exons 14 in *USO1*, and 4/3b in *RAC1* (Figure 3c and Appendix A).

### 2.4. Isoform-Switching Analysis Confirms the Observed ESRPs Modulation of AS in MCF-7 Cells

To further evaluate the AS changes induced upon ESRP1/2 combined silencing, an isoform-based approach was applied using the IsoformSwitchAnalyzeR algorithm [28]. This analysis identified isoform switching events driven by changes in the relative abundances of isoforms, referred to as isoform differential usage. This analysis evidenced 814 significant isoform switching events involving 567 genes (Appendix A). These genes were functionally related to actin cytoskeleton organization (e.g., *MYO1B*, *FLNB*, *MYO6*, *RAC1*), tissue morphogenesis (e.g., *CD44*, *FGFR2*, *SCRIB*, *ARHGAP35*, *MYO9A*, *NF2*), and regulation of cell adhesion (e.g., *BMP7*, *CD44*, *DLG1*, *DNM2*, *DUSP3*, and *FLNA*) (Appendix A).

The identified isoform switching events resulted in significant downstream consequences that distinguish the induced from repressed isoforms. Notably, switching isoform pairs differ by 3′UTR and 5′UTR lengths, by type and number of protein domains, IR events, sensitivity to NMD, and their coding potential. In particular, the combined silencing of ESRP1/2 resulted in the significant enrichment of isoforms characterized by longer 3′UTRs (q-value = 2.5 × 10^−5^) and longer 5′UTR (q-value = 0.05), with protein domain gains (q-value = 4.6 × 10^−5^), more IR events (q-value = 7.8 × 10^−3^), and more NMD-insensitive (q-value = 0.02) (Appendix A). Comparing the proportion of isoforms with opposite features showed that these events are driven mainly by single ES events, followed by alternative splice site usage, and differential usage of transcription start and termination sites (Appendix A).

The genes involved in the most significant switching events (*USO1*, *FLNB*, *SLK*, *TUFT1*, *ENAH*, *RAC1*, *UAP1*, *MYO1B*, and *EXOC7)* are also reported among those with the most significant differential ASEs by rMATs analysis. In total, 172 genes were confirmed as altered by both analyses, including 148 (26.1%) genes with ES events, 35 (6%) genes with MXE events, 10 (2%) genes with A5′SS events, 8 (1.40%) genes with A3′SS events, and 4 genes (0.70%) with RI events (Appendix A). The isoform switching event involving the *SLK* gene is reported in (Appendix A).

### 2.5. Network-Based Functional Prediction of ASEs upon ESRP1/ESRP2 Silencing

To get insights into the downstream consequences of the ESRP1/2-modulated ASEs at the protein isoform level in BC, we performed a functional protein–protein interaction (PPI) network analysis. This analysis was achieved considering the protein domain–domain interactions (DDIs) that were predicted to be enriched/depleted by the identified ASEs in our dataset.

The exploration of public PPI and DDI data enabled the construction of a global network for MCF-7 cells, containing 1982 nodes and 3627 interaction edges (Supplementary Materials Table S7a). The network was characterized by a scale-free structure and an average degree of four (range = 1–209) (Appendix A). To decipher interactions that are potentially regulated by ESRP1/2 silencing, the analysis was then restricted to a subnetwork of 95 genes which harbor significant ES events and whose inclusion/exclusion levels correlated (*p* < 0.05) with *ESRP1* and *ESRP2* mRNA levels in ERα+ BCs (Appendix A). Furthermore, a differential Interaction Score (dIS) was computed based on the relative abundances of AS isoforms to evaluate the effects of ASEs on DDIs and PPIs. The resulting weighted network was composed of 410 nodes and 490 interactions (Appendix A). The GO enrichment analysis of these nodes revealed an enrichment in cell-signaling-related terms such as *protein phosphorylation*, *regulation of small GTPase mediated signal transduction* and cell movement, the *actin filament-based process*, *regulation of cell projection organization*, *cell morphogenesis*, *regulation of cell adhesion*, but also cell division and RNA splicing (Appendix A).

The analysis of the network highlighted biologically relevant hub nodes. For instance, the cell signaling GTPase, *RAC1*, represented the main network hub, with a centrality degree of 80 (Figure 4a), with 63 (79%) interactions predicted to decrease (dIS < 0) following the skipping of exon 4/3b upon ESRP1/2 silencing (Appendix A). RAC1-KRAS is the most decreasing interaction (dIS = −1.22), followed by the one with another Rho GTPase, ARHGAP33 (dIS = −0.89). Notably, a functional enrichment analysis of RAC1 interacting partners showed an enrichment in *regulation of small GTPase-mediated signal transduction*, *regulation of GTPase activity*, *cell morphogenesis*, the *actin filament-based process*, and *cell junction assembly* (Appendix A). Other depleted interactions include a self-interaction of USO1 (dIS = −1.96), involving the globular p115 head domain encoded by the skipped exon 14. Similarly, SCRIB was involved in 14 interactions, of which 9 (64%) involving its PDZ domain were predicted to decrease (dIS < 0) (Figure 4b). In contrast, the remaining five involving the Leucine-rich repeats domain (LLR_8), were predicted to increase (dSI > 0). Notably, the *SCRIB* gene had a significant isoform switching event upon ESRP1/2 silencing, leading to significant repression of the shortest isoform (SCRIB-207), annotated only for the PDZ domain, while induction of the main longer isoform (SCRIB-201) also annotated for the LLR_8 domain (Appendix A).

### 2.6. ESRP1/2-Regulated ASEs Occur upon ERα Silencing in Hormone-Deprived MCF-7 Cells

Since apoERα regulates *ESRP1* and *ESRP2* expression, the overlap of ESRP1/2-modulated ASEs with those observed upon apoERα silencing in MCF-7 cells was evaluated [21]. The overlap revealed 64 commonly dysregulated events: 50 ES (78%), 5 MXE (7.80%), 4 A5′SS (6.25%), 2 A3′SS (3.13%), and 3 (4.68%) RI events (Appendix A). The dPSI of these ASEs was correlated (rho = 0.55, *p* < 0.0001) between the two experiments, particularly for ES (rho = 0.89, *p* < 0.0001), and showed coherent regulation between both experiments (Appendix A). The top four overlapping and coherent ASEs were the ES of *USO1* exon 14, the ES of *APLP2* exon 8, the inclusion of exon 12 of *SCUBE2*, and the inclusion of exon 17 of *MYOF* (Appendix A). Notably, the RBP motif analysis on the overlapping ASEs indicated the presence of ESRP1/2-binding motifs within the regions involved in these ASEs.

Furthermore, the overlapping ASEs between apoERα and ESRP1/2 datasets were investigated in BC samples. Specifically, their inclusion levels (PSI) were retrieved from the SpliceSeq data, observing 27 (42%) ASEs detectable in primary BC samples (e.g., 773 ERα+, and 192 ERα−), and in a subset of 113 normal breast tissues (Appendix A). Among them, 23 (85.2%) and 18 (66.7%) significantly correlate (*p* < 0.05) with *ESRP1* and *ESRP2* mRNA levels in these tumors, respectively, and 12 ASEs correlate with gene expression (Figure 5a and Appendix A). The most significantly correlated ASE is an ES involving *USO1* (rho = 0.44 and 0.45 for *ESRP1* and *ESRP2*, respectively; *p* < 0.0001). The top three correlated ASEs with *ESR1* expression are an ES involving *SPTAN1* (rho = 0.39, 0.19, and 0.42 for *ESRP1*, *ESRP2*, and *ESR1*, respectively; *p* < 0.0001), an ES in *MYOF* (rho = −0.30, −0.23, and −0.36 for *ESRP1*, *ESRP2*, and *ESR1*, respectively; *p* < 0.0001), and an ES involving *VPS39* gene (rho = 0.34, 0.19, and 0.23 for *ESRP1*, *ESRP2*, and *ESR1*, respectively; *p* < 0.0001) (Figure 5a).

Furthermore, the 27 overlapping ASEs were analyzed in a siESRP1 RNA-Seq experiment in endocrine-resistant BC cell lines (2C3, 9C3 from [17]), showing 14 ASEs (63.6%) were significantly (*p* < 0.05) regulated upon *ESRP1* silencing (Figure 5b and Appendix A). The dPSIs of these ASEs are coherent between our siESRP1/2 silencing experiment and the one performed in 2C3 (rho = 0.61, *p* < 0.05), and in 9C3 cells (rho = 0.65, *p* < 0.05).

To validate these results in primary tumors, the differential inclusion/exclusion levels of the 27 overlapping ASEs were evaluated in TCGA BCs relative to the median mRNA expression levels of *ESPR1*, *ESRP2*, and *ESR1* (Figure 5c and Appendix A). This analysis confirmed a significant correlation between the levels of inclusion of these 27 ASEs and the mRNA levels of *ESRP1* (rho = 0.83, *p* < 0.0001) and *ESRP2* (rho = 0.65, *p* < 0.001) in these tumors. In addition, the silencing of ESRP1/2 in MCF-7 mirrored the changes observed between tumor samples with high versus low expression of the two genes. Furthermore, when BC PSI data were compared with PSI values of normal breast tissues, the levels of 21 ASEs resulted significantly different between the two groups (Figure 5c and Appendix A). In particular, the most significant ASE is an ES of the *MYOF*, whose exon 17 was significantly more excluded in BC compared to normal tissue (dPSI = −0.33, *p* < 0.0001) (Figure 5d). Conversely, the exon 23 of *SPTAN1* is significantly more included in BC compared to normal tissue (dPSI = −0.16, *p* < 0.0001). Coherently with the analysis of clinical data in relation to the *ESRP1* and *ESPR2* expression, most of the 27 ASEs were significantly related (*p* < 0.05) to the tumor molecular subtype (22 events), the fraction of altered genome (20 events), the menopause status, and the diagnosis age (8 events) (Appendix A).

To decipher which core molecular pathways are associated with the AS changes observed upon ESRP1/2 silencing, a correlation analysis between the inclusion level (PSI) of each event in ER+ BCs and molecular pathways was performed using the PEGASAS algorithm [29]. This pathway-guided enrichment analysis revealed a differential enrichment of ASEs in two clusters of molecular pathways, the first corresponding to inflammation and EMT-related pathways and the second mainly to proliferation and cell-cycle-related pathways (Appendix A). Focusing on the 27 overlapping ASEs, the analysis resulted in the identification of two clusters of molecular pathways that were significantly correlated (Figure 5e). Specifically, the first cluster was composed mainly of terms related to EMT and inflammation, including *interferon gamma response*, *TNFA signaling* via *NFKB*, *and IL6*/*JAK*/*STAT3 signaling*, and it was positively correlated (r > 0.3) with six ES events involving *MYOF*, *SULF2*, *MYL6*, TBC1D13, ZNF267, and *SCRIB*. Conversely, the second cluster of molecular pathways related mainly to cell proliferation, such as *G2M checkpoints* and *E2F targets,* and it was correlated with ES events at *USO1*, *PTPRF*, *NF2*, *MYOB*, *SCUBE2*, *VPS39*, and *SPTAN1* (Figure 5e and Appendix A).

## 3. Discussion

In this study, we explored the functional roles of the two key epithelial splicing regulators, ESRP1 and ESRP2, in the control of ERα+ BC transcriptome. The direct ERα regulation of these gene expression was provided by RNA-Seq and ChIP-Seq data analysis in MCF-7 and primary tumors. This is further supported by the correlation analysis performed in primary tumor data and by the strong *ESRP1* and *ESRP2* downregulation upon ERα silencing in hormone-deprived MCF-7 [21]. Whereas a clear ERα chromatin binding is observed at the *ESRP2* promoter and first intron in both cell lines and primary BCs, the binding at *ESRP1* promoter was not observed in our hormone-deprived MCF-7. This suggests the existence of secondary factors downstream of ERα signaling contributing to ESRP1 downregulation upon ERα silencing. Instead, ERα binding was observed at the *ESRP1* promoter in primary BCs, resistant or not to endocrine treatment. In addition, the depletion of ERα and ESRP1/2 in MCF-7 revealed commonly regulated ASEs, strongly correlated with pathways that may contribute to BC development and progression.

ESRP1 and ESRP2 are not only overexpressed in ERα+ BCs, but also show significant copy number alterations in these tumors. Indeed, we observed *ESRP1* amplification in 60% of the analyzed samples, in line with [30], showing that *ESRP1*, among other genes located near the 8q24 amplicon, is overexpressed in BC and its overexpression relates to copy number gains in these tumors. This evidence was also confirmed in MCF-7 by the analysis of copy number alterations from the Cancer Cell Lines Encyclopedia [31] showing the amplification of *ESRP1* but not of the *ESRP2* gene (Appendix A). In addition, among the 51 BC cell lines with copy number alteration data in this database, 32 were associated with an ESRP1 amplification (62.54%). Only in 14 cell lines was ESRP2 amplified (27.45%).

In our data, ESRP1/2 silencing altered the expression of almost one thousand genes, including several interferon-responsive genes such as *IFI27* and *IFI6*, whose expression is associated with reduced OS in BC patients [32,33]. Specifically, *IFI27* expression is induced by estradiol, and it has been identified as an ERα-associated protein in different BC cell lines [33]. *IFI6*, whose levels are reduced by ESRP1/2 depletion, has been reported as overexpressed in poor-prognosis BC, and it promotes the metastatic potential of BC cells [34]. Other interferon-inducible genes repressed in our data include *OS1*, *OS2*, *OS3*, *OASL*, and their upstream regulators, *STAT1* and *DDX60. STAT1* and *DDX60* expression increases in tamoxifen-sensitive and -resistant MCF-7 cell lines [32]. These data highlight a role of ERα-ESRPs axis in regulating interferon signaling at different levels and suggest novel insights into the association of ESRP1 overexpression and endocrine resistance in BC.

Notably, although ESRPs are well described for their role in controlling EMT, their silencing in MCF-7 did not induce a strong overall gene expression change of EMT-related genes, in line with previous studies [13,17]. Conversely, the ESRP1/2 silencing extensively influenced the AS pattern of these genes. Indeed, our analysis of ESRP1/2-modulated ASEs confirmed most of the EMT-related AS signatures reported in other studies [17,35,36]. Notably, comparing the ASE in our cellular model with those in a study by Yang and colleagues [15], 109 coherent ASEs were found to be common in both studies (Appendix A). In addition, comparing our ASEs with those identified in an EMT time-course RNA-seq dataset of *ZEB1* overexpression in epithelial cells [15] revealed 171 common ASEs, most of which (155 ASEs, 90.64%) showed coherent changes with our study (Appendix A). Among the ASEs harboring a predicted ESRPs binding motif, 33 interactions were supported by CLIP-seq data from [37]. These ASEs were found to be tightly regulated by ESRP1 and hnRNPM during EMT [37].

Importantly, the inclusion/exclusion levels of ESRPs-modulated ASEs were confirmed to be related to *ESRP1* and *ESRP2* mRNA levels in primary ERα+ BC from TCGA. These results support the direct activity of these proteins in both in vitro models and primary tumor cells. Furthermore, this analysis revealed ASEs with potential prognostic value in BC. Interestingly, 11 ESRPs-modulated ASEs (at *ANXA6*, *AP1B1*, *CTNND1*, *DNM2*, *ENAH*, *FNBP1*, *MBNL1*, *SLK*, *SPAG9*, and *TSC2*) were recently identified as a robust classifier AS signature of Basal-like BCs, into A and B subtypes, correlating with the tumors’ aggressiveness and responsiveness to chemotherapy [38].

In addition, we performed an unbiased prediction of ESRP1/2 silencing effects on protein–protein interaction networks in MCF-7. The network formalism has proven to be useful for exploring the predicted effects on the proteome network, providing powerful insights into the most affected interactions in our dataset. For example, focusing on a subnetwork of genes harboring ASE correlated with ESRP1/2 mRNA levels allowed us to highlight *RAC1* as the main network hub. *RAC1* was characterized by the highest number of decreasing interactions, all involving its Ras family domain. The ASE on *RAC1* is the exclusion of exon 3b/4 which belongs to the Rac1 isoform known as Rac1b. The Rac1b isoform is constitutively active and highly expressed in different cancers, including breast, thyroid, colorectal, and lung tumors [39,40,41]. Furthermore, the characterization of the canonical Rac1 and Rac1b isoforms using data from BC cell lines, including MCF-7 cells, indicated that the two antagonistically act on EMT. Canonical Rac1 induces EMT, while Rac1b isoform, which is highly expressed in epithelial cells, represses the process. Rac1b expression is lost in undifferentiated BC tissues and mesenchymal cell lines [42,43]. The functional correlation analysis between the inclusion levels of the AS exon 4 (or, canonically, exon 3b) in ERα+ BCs and signaling pathways indicated an anticorrelation with pathways such as PI3K/Akt/mTOR signaling pathways, metabolism, and oxidative phosphorylation with TGFB signaling and EMT, in line with previously reported results [44,45]. In contrast, higher inclusion levels of exon 4/3b (and thus, higher Rac1b isoform expression) correlated with estrogen response and Myc targets pathways, in line with [43], showing a clear association between higher Rac1b/Rac1 ratio and ERα+ epithelial BC phenotype.

Our functional network analysis also reported the scribble planar cell polarity protein (SCRIB) as a network node characterized by decreasing interactions, consistently with a significant switching event involving the main longer and shorter isoform. SCRIB has already been described as regulated by ESRP1 in ERα+ BC [17], but is also closely implicated in epithelial cell polarity and EMT [46]. SCRIB is crucial for E-cadherin-mediated cell–cell adhesion and for mitotic spindle orientation in epithelial cells [47]. Our differential AS analysis indicates that ESRP1/2 silencing results in significant AS regulation of SCRIB, with increased expression of exon 16 instead of exons 17 and 18. A recent study by [48] investigating the post-transcriptional regulation of *SCRIB* reported a preferential usage of putative exons in BCs, including the increased inclusion of exon 16 in ERα+ with respect to ERα− tumors. The PDZ domain coded by this exon, characterized by the lowest dIS, interacts with the subunit beta and alpha of serine/threonine protein phosphatase PP1. SCRIB and PP1 form a complex with leucine-rich repeat protein SHOC2 downstream of MRAS, which is involved in the regulation of ERK pathway dynamics and cell polarity [49]. In addition, our isoform switching and network analysis are in line with [48] showing that *SCRIB* transcripts expressing the conserved LL8_R N-terminal domains, induced in our dataset, are preferentially overexpressed in BCs, while the transcripts only expressing the PDZ C-terminal domains are lost in these tumors. Notably, the case of *SCRIB* AS regulation in our dataset highlights the strength of our network-based approach when it comes to combining both differential local and isoform-based AS analyses. Indeed, differential AS analysis by rMATS indicate changes in the single disjoint exons, which are most often difficult to decipher in a biological interpretation. Instead, the network was designed by integrating rMATS data with isoform-based results from IsoformSwitchAnalyzeR and allowed the identification of significant changes at exons encoding for protein domains that are missed by local AS analysis.

Another ASE experimentally confirmed in our datasets and whose consequence was predicted in our network is the skipping of *USO1* exon 14. This exon encodes for a protein domain involved in the USO1 dimerization, the interaction with Rab1, and its recruitment on COP II coated vesicles [50].

As an effect of the ESRP1 and ESRP2 downregulation, the silencing of ERα in hormone-deprived cells mirrored the silencing of ESRP1/2 on a specific subset of ASE [20]. Specifically, 64 ASE showed the same regulation in both silencing experiments, among which 27 ASE were validated in primary ERα+ BC data. Notably, the ES events at *USO1*, *PTPRF*, *NF2*, *MYOB*, *SCUBE2*, *VPS39*, and *SPTAN1* genes positively correlated with the cell proliferation process, and their inclusion levels correlated with *ESRP1*, *ESRP2*, and *ESR1* expression. A second splicing pattern potentially involving the ERα–ESRPs axis includes ES events of *MYOF*, *SULF2*, *MYL6*, *TBC1D13*, *ZNF267*, and *SCRIB* genes positively correlated with interferons signaling pathways, metabolism control, and the EMT process. Furthermore, several ER-Golgi trafficking signature genes which promote BC metastatic progression were also differentially spliced in both silencing experiments [51].

Other ASE patterns overlapping between the two silencing experiments but not present in the SpliceSeq data include the ES event at the amyloid precursor-like protein 2 encoding gene (*APLP2*). The exon involved in this event encodes for a protein domain essential for the interaction with *KLK2* encoded proteins, which in turn is found aberrantly expressed and is used as a prognostic biomarker in many cancer types, including BCs [52].

Our data suggest that ERα-dependent transcription of ESRP1 and ESRP2 controls several individual mRNA splice isoforms that correlate with ERα+ BCs development and progression. Nevertheless, our data of course do not address the individual contribution of either ESRP1 or ESRP2, which would require individual CRISPR-Cas9, as well as re-transfection of these factors in void cells, as previous studies in our and other labs have suggested. The analysis of ESRP1/2 in MCF-7 has given important insights and significant correlation with tumor-observed analysis. Other BC cell lines may thus provide more important information.

## 4. Materials and Methods 

### 4.1. Analysis of ESRP1, ESRP2, and ESR1 Expression in TCGA Clinical Data

Analysis of *ESRP1* and *ESRP2* expression in primary tumor RNA-Seq data was performed by considering the 774 BCs from the BRCA cohort of TCGA. All 774 samples were only ERα+ tumors, as defined based on the immunohistochemistry level of ERα. The *ESRP1*, *ESRP2*, and *ESR1* expression levels in Fragment Per Kilobase Mapped Reads (FPKM) were retrieved from the GDC (Genomic Data Commons) data portal [53]. Clinical data were obtained from cBioPortal [54], considering the clinical data from the dataset named “Breast Invasive Carcinoma (TCGA, PanCancer Atlas)”. The same dataset was used to obtain the copy number status information expressed as the GISTIC score. Samples with a positive ERα status based on immunohistochemistry but associated with an *ESR1* FPKM lower than 1 were excluded from the analysis. Meanwhile, 192 BCs characterized by a negative ERα immunohistochemistry level and *ESR1* FPKM lower than 1 were considered ERα negative tumors.

The statistical analysis of *ESRP1*, *ESRP2*, and *ESR1* expression in relation to different patient clinical data was performed considering both the expression level of these genes and by separating the samples in highly and lowly expressed patients based on the median expression values. The analysis of the gene expression levels in relation to continuous clinical data was performed using a Spearman correlation analysis (implemented in the *cor_test* R function). Gene expression differences in patients stratified by categorical variables were analyzed using a Kruskal–Wallis test (implemented in the *Kruskal.test* R function). Chi-square analysis was applied to analyze categorical clinical data with respect to two groups of patients (high and low) defined based on the median expression of the analyzed genes.

### 4.2. Overlap with ERα ChIP-Seq Data

Analysis of ERα chromatin binding at *ESRP1* and *ESRP2* gene locus was performed by considering the reference of ERα chromatin interactions in MCF-7 cells from [22]. In the analysis, we considered data of hormone-deprived cells (E2-independent), cells maintained in hormone-enriched media (E2-constitutive), or cells treated with 17-beta estradiol (E2-treatment). ERα chromatin bindings in tamoxifen-sensitive or resistant cell lines (MCF-7 and BT-474) and primary BCs were obtained from GSE32222 [23]. The peak genomic coordinates were converted in hg38 genome assembly using LiftOver (https://genome.ucsc.edu/cgi-bin/hgLiftOver (accessed on 1 June 2020).

### 4.3. Cell Culture and siRNA Transfection

MCF-7 cells were routinely grown in DMEM (Dulbecco’s Modified Eagle’s Medium) (Life Technologies, 31053-028, Waltham, MA, USA) supplemented with 10% heat-inactivated FBS (fetal bovine serum) (Euroclone S.p.A, ECS0180L, Milan, Italy) and 2 mM L-glutamine (ThermoFisher Scientific, 25030-024, Waltham, MA, USA). Batches of human cell lines were purchased from ATCC (American Type Culture Collection). Cells were cultured at 37 °C with 5% CO_2_. The cell transfection in suspension was performed by seeding one million cells and a transfection mix composed of siRNAs (30 nM final concentration) and Lipofectamine3000 (ThermoFisher Scientific, L3000015, Waltham, MA, USA), as transfection reagents. Cells were left to attach overnight in the cell incubator and, subsequently, the medium was refreshed. Two custom-designed siRNAs were used to target ESRP1 and ESRP2 (siESRP1 5′-CAGCAGGUGCUGAAUCGUUCUCCU-3′, siESRP2 5′-GCACAUCACUAGAGGUGGCUCGUUU-3′); stealth RNAi™ siRNA Negative Control medium GC was used as a control (siCTRL). All the siRNAs were purchased from Thermo Fisher Scientific (Waltham, MA, USA).

### 4.4. RNA Isolation, RT-PCR, RNA-Seq Libraries Preparation and Sequencing

Forty-eight hours after siRNAs transfection, total RNA was isolated from MCF-7 cells using the RNeasy Mini Kit, according to the manufacturer’s protocol (Qiagen, 74104, Hilden, Germany). Experiments were performed in three biological replicates. First-strand cDNA synthesis was performed with the SensiFAST™ cDNA Synthesis Kit (Bioline, BIO-65054, London, UK). Qualitative PCR analysis was performed using MA038 EconoTaq 2xMaster Mix (Lucigen, Middleton, WI, USA). PCR products were analyzed on a 3% agarose gel, stained with SYBR™ Safe DNA Gel Stain (S33102 ThermoFisher Scientific). The list of primers (purchased from Eurofins Scientific, Luxemburg) that were used for the validation of the selected ASEs is presented in Table 1.

**Table 1 ijms-23-07835-t001:** Primers used for the qPCR validation of target ASEs.

Gene	Target ASE	Forward Primer	Reverse Primer
RAC1	ES_03_**04**_05	5′-GACAGATTACGCCCCCTATCC-3	5′-CAGGACTCACAAGGGAAAAGC-3′
SCRIB	ES_16_**17**_18	5′-CATCCGCAAGGACACACCT-3′	5′-CCTTATAGGGTGTGGAGCCCT-3′
MYOF	ES_16_**17**_18	5′-CTCTGGTGGGGAAGTGGAAG-3′	5′-CGTGTACTCTCTGGGGCTTC-3′
USO1	ES_13_**14**_15	5′-TGCTCAGGGTTCAACTTGCT-3′	5′-GGGACAATTGCTTAGCCAGG-3′

RNA quality check (RNA integrity number (RIN) > 8) was achieved with a Fragment Analyzer (Advanced Analytical Technologies, Inc., Ankeny, IA, USA) and quantified with Qubit (Qubit™ RNA HS Assay Kit, ThermoFisher Scientific, Waltham, MA, USA; Q32852). Total RNA, depleted of ribosomal RNAs fractions, was used to construct the RNA-Seq libraries using the Illumina TruSeq RNA sample prep kit (TruSeq™ RNA Sample Prep Kit v2-Set B, Illumina, RS-122-2002, San Diego, CA, USA). The 75 nt paired-end (PE) cluster generation was performed using cBot on Flow Cell v3 (TruSeq PE Cluster Kit v3-cBot-HS, Illumina, PE-401-3001, San Diego, CA, USA). The sequencing of libraries was performed on the HiSeq sequencing system (Illumina, San Diego, CA, USA). The raw RNA-Seq data were deposited in Gene Expression Omnibus (GEO) with the identifier GSE206474.

### 4.5. Protein Extraction and Western Blot

Whole-cell lysate was harvested in boiling lysis buffer (25 mM Tris·HCl pH 7.6, 1% SDS, 1 mM EDTA, 1 mM EGTA) and 50 µg of total protein extract was loaded into an 8% acrylamide gel. The antibodies used were designed against ESRP1(Sigma-Aldrich, Milan, Italy) [55], ESRP2 (abcam ab113486), GAPDH (Santa Cruz Biotechnology, sc-32233, Dallas, TX, USA), and HSP90 (abcam ab59459).

### 4.6. Differential Expression and Differential Alternative Splicing Analysis

Raw reads were assessed for Phred quality scores using FASTQC (https://www.bioinformatics.babraham.ac.uk/projects/fastqc/ (accessed on 1 January 2021), and low bases and adaptor sequences were trimmed off using Fqtrim (http://ccb.jhu.edu/software/fqtrim/ (accessed on 1 January 2021) retaining only reads of 75 bass length. Then, clean reads were aligned against the human reference genome (GRCh38.p10) with Gencode v27 annotation using STAR v2.5.1b [56]. STAR was run in a two-pass mode allowing alignments to the transcriptome coordinates by setting the option—quantMode to TranscriptomeSAM. The expression levels in read counts, TPM, and FPKM units were then estimated at both genes and isoforms levels by running RSEM [57] on the alignment files in default parameters.

### 4.7. Differential Expression Analysis

Differentially expressed genes and isoforms upon ESRP1/2 silencing were identified using the DESeq2 R package (v1.26.0) in default parameters [58]. The expression at the isoform level was summarized to the gene level using the *tx-import* Bioconductor package [59], and the resulting count matrices were provided to DESeq2. Prior to DE analysis, lowly expressed genes and isoforms were discarded from the analysis, and only genes or isoforms with more than 10 normalized read counts in at least one condition (3 out of 6 samples) were considered for further downstream analysis. A gene or isoform was labeled differentially expressed if its associated BH-adjusted *p*-value < 0.05. All the data visualization plots, including heat maps, volcano plots, and MA plots were made using the ggplot2 R package (v.3.2.1) [60].

### 4.8. Gene Ontology Enrichment Analysis

Gene Ontology terms enriched for upregulated and downregulated genes were obtained using the Gene Annotation and Analysis Resource Metascape program [61]. The lists of upregulated and downregulated genes were analyzed separately using the Single List Analysis option. The statistically enriched GO terms were obtained from the GO Biological Processes. Only GO terms that were associated with an enrichment factor > 1.5 and an accumulative hypergeometric test adj. *p*-value < 0.05 were significant. To reduce redundancy, the GO terms showing a high number of overlapping genes and a large degree of redundancies were clustered into groups based on their degree of similarities, and each group or cluster was represented by the top significant GO term. The top 20 significant clusters were selected for visualization purposes.

### 4.9. Isoform Switching Analysis

To test for isoform switching events, the IsoformSwitchAnalyzeR tool was applied [28]. Briefly, from the RNA-seq data, the tool takes as inputs isoforms expression levels quantified in TPM (transcript per million fragments mapped) units normalized to transcript length and then calculates an isoform fraction (IF) ratio by dividing the isoform expression with the expression of the parent gene (TPM_iso_/TPM_gene_). Lowly expressed genes with less than 1 TPM and lowly expressed isoforms that do not contribute to the expression of the gene (IF < 0.01) were excluded from downstream analysis. The IF was then calculated per each of the remaining isoforms and conditions. A dIF (IF_silencing_–IF_control_) representing the difference in isoform usage between the two conditions was calculated. A cutoff criterion was applied by selecting only those isoforms for which ESRP1/2 silencing induced a significant change (BH-corrected *p*-value < = 0.05) in IF by at least 10% (i.e., |dIF| > 0.1). Next, the sequences corresponding to those isoforms showing significant switching events upon ESRP1/2 silencing were extracted and then annotated for the presence of signal peptide sequences, coding potential, and for their associated pfam protein domains using signalP [62], CPC2 [63], and Pfam [64] tools, respectively. The biological consequences of the observed switches, including intron retention, domain gain/loss, coding/non-coding potential, and shortening/lengthening of the open reading frame were then evaluated for the switching isoforms from the same parent gene. Next, according to the applied annotation on the switching isoforms, genes were classified into genes with or without downstream functional consequences.

### 4.10. Differential Alternative Splicing Analysis

The list of differentially regulated AS events upon *ESRP1*/*2* silencing was identified using rMATS [24,25]. rMATS is an improved version of the original MATS method [65], and is described to provide a 100-fold increase in running time compared to older versions of MATS. rMATS takes as input a gene annotation model file together with the list of RNA-seq samples, either in fastq or bam format, and provides the inclusion ratio of each specific genomic region included in the annotation. rMATS provides a splice index (PSI) as a measure of the inclusion level of each event by calculating the ratio of the paths supporting the inclusion of the alternative exon or splice site divided by the total number of the paths supporting both inclusion and exclusion of exon. rMATS quantifies changes in the five major splicing events, including single exon skipping (ES) events, alternative splice sites (A5′SS and A3′SS), mutually exclusive exons (MXE), and intron retention (RI) events. All the sequences and annotations used in this analysis were based on GRCh38 genome assembly and Gencode v27 annotation. To ensure quantification of expressed events, a prefiltering criterion was applied by only considering those splicing events whose supporting reads are at least 10 in at least two samples per condition. In addition, splicing events with a ΔPSI value between the silencing and control conditions less than 10% (|ΔPSI| < 0.1), or associated adj *p*-value > 0.05, were excluded from the downstream analysis.

### 4.11. RBP Binding Motif Enrichment Analysis

To identify RNA-binding proteins as putative regulators of the observed changes in each splicing event identified, the sequences of the regulated DS events extended ±200 nucleotides on both sides were scanned for the occurrence of RBP binding motifs. The RNA binding motifs for 105 different splicing factors collected from the RNAcompete study [66] were used to perform binding motif enrichment analysis. For several RBPs, the motif from different species was confirmed in a previous study [30] to be conserved between the human and other species. This includes RBM47 (chicken), SF1 (Drosophila), SRP4 (Drosophila), TRA2 (Drosophila), and PCBP3 (mouse). Next, the MoSEA (Motif Scan and Enrichment Analysis) package was used to search the sequence of the splicing events for the occurrence of RBP binding motifs [30]. The tool FIMO (Find Individual Motif Occurrences) [67] was used to scan the sequences of the events for the presence of the RBP motifs using a *p*-value < 0.001 as a cutoff. The binding motif enrichment was performed by comparing the number of occurrences of the binding motifs of the RBPs in the regulated events with those observed in a pool of 100 randomly selected sequences of the same size from equivalent regions in non-regulated events (|ΔPSI| < 0.01 and *p* < 0.5). Motif enrichment was performed separately for the two directions of splicing changes (ΔPSI > 0.1 or ΔPSI < −0.1). An enrichment z-score per RNA binding motif, region, and direction of regulation was calculated by normalizing the observed frequency in the regulated events set with the mean and standard deviation of the 100 random control sets. The 100 random control sequences were sampled from non-regulated events for each region of regulation. An RBP was considered enriched if associated with a (z-score > 1.96). The obtained z-scores per binding motif, region, and event were then visualized using the ggplot2 Bioconductor package [60].

### 4.12. Overlap with ASEs in Primary Tumor Data

Analysis of ASEs identified as dysregulated in both siESRP1/2 and siERα experiments was performed, considering the annotations from SpliceSeq [27]. This database reports the PSI values of different ASEs detected in the RNA-Seq data from TCGA. Specifically, the analysis was performed by retrieving all the PSI values of the TCGA BRCA cohort from the database website (http://projects.insilico.us.com/TCGASpliceSeq/ (accessed on 1 June 2021). Then, the ASE coordinates from the database were converted to hg38 assembly using the Liftover tool (https://genome.ucsc.edu/cgi-bin/hgLiftOver (accessed on 1 June 2021) and overlapped with the ASE coordinates from the rMATS analysis. Spearman correlation analysis was performed to evaluate the relationship between *ESRP1* or *ESRP2* expression in FPKM with the PSI value associated with each ASE. A Wilcoxon Rank-Sum test was performed to evaluate the difference in PSI value distributions between data from patients divided based on *ESRP1* or *ESRP2* expression levels, between ERα+ and ERα− BCs, or between ERα+ BCs and normal breast tissue. Correlation analysis between the ASE inclusion/exclusion levels and signaling pathways from the Molecular Signatures Database (MSigDB) hallmark gene sets was performed using PEGASAS in default settings [29].

### 4.13. Definition of Domain–Domain Network for ASE Functional Prediction

The construction of the protein interaction network characterized at the domain level was divided into two steps: (i) the exon and isoform annotation with protein feature into a single data structure, and (ii) the actual network generation of interactions and associated scores.

The annotation of the exons of isoforms expressed in the dataset (median isoform expression > 1 TPM in both conditions) was performed using protein domain information from the Ensembl via the AnnotationHub v2.22.1 and EnsemblDB v2.14.1 R packages. The annotated isoforms were joined with results from IsoformSwitchAnalyzeR and rMATS analysis. Only ES events were considered for the analysis. Then, protein domain annotation was filtered considering only spliced exons annotated with Pfam domain to avoid redundancies. This annotation step resulted in a dataset of 8284 protein-coding genes (20,819 isoforms and 34,001 exons), coding for 20,819 proteins containing a total of 3854 protein domains.

Generation of the domain–domain interaction network was performed in a two-step process. In the first step, all pairwise protein interactions at the domain level for all annotated isoforms were reconstructed using protein and domain interactions data from BioGRID v4.4.199e [68] and 3did vMar_4_2021 [69]. Interacting proteins that were annotated with at least a pair of interacting domains were considered as interacting. Once the pairwise interaction table was built, interactions involving only genes characterized by a significant ES event or isoform switch were kept.

In the second step, an Interaction Score (IS) score for each interaction was computed. For each pair of interacting isoforms, the IS is the product of the sums of IFs for each isoform, predicted from the IsoformSwitchAnalyzeR analysis, to contain the domain(s) involved in the interaction. 

Finally, the change of each interaction between two conditions was evaluated by computing the differential Isoform Score (dIS), defined as the logarithm in base two of the ratio between the IS computed in the two compared conditions.

The network was visualized with Cytoscape v3.8.2 [70]. The node size was set proportional to the node degree, while the links width was set be proportional to the absolute dIS. The link colors were set equal to the dIS, using a blue-to-red palette (blue colors for negative values and red for positive ones). Isolation of the subnetwork was achieved by extracting overlapping genes of our dataset affected by a significant ES event and ASEs data from TCGA, as previously described in the Method Section 4.10.

## 5. Conclusions

ESRP1 and ESRP2 are the core factors responsible for maintaining an epithelial phenotype of BC. Several aspects related to these factors, including the expression status, copy number variations, as well as their activities in controlling the ERα+ BC transcriptome, were extensively explored in this study. Our data clearly indicate a significant correlation between *ESRP1*/*2* and *ESR1* mRNA levels both in MCF-7 and primary tumors. In addition, the copy number amplification of the *ESRP1* gene locus is associated with concomitant mRNA overexpression in ERα+ BCs and is associated with poor prognosis in these tumors. The gene expression changes caused by ESRP1/2 depletion in MCF-7 indicate that the two factors regulate the expression of genes involved in the inflammatory response, previously reported as a molecular pathway of the onset of endocrine resistance. The analysis of AS changes indicates that ESRP1 and ESRP2 control the expression of exons that are associated with specific tumor molecular features and patient clinical outcomes. Collectively, our data suggest that the Erα–ESRP axis is pivotal for the onset and progression of BC.

## Figures and Tables

**Figure 1 ijms-23-07835-f001:**
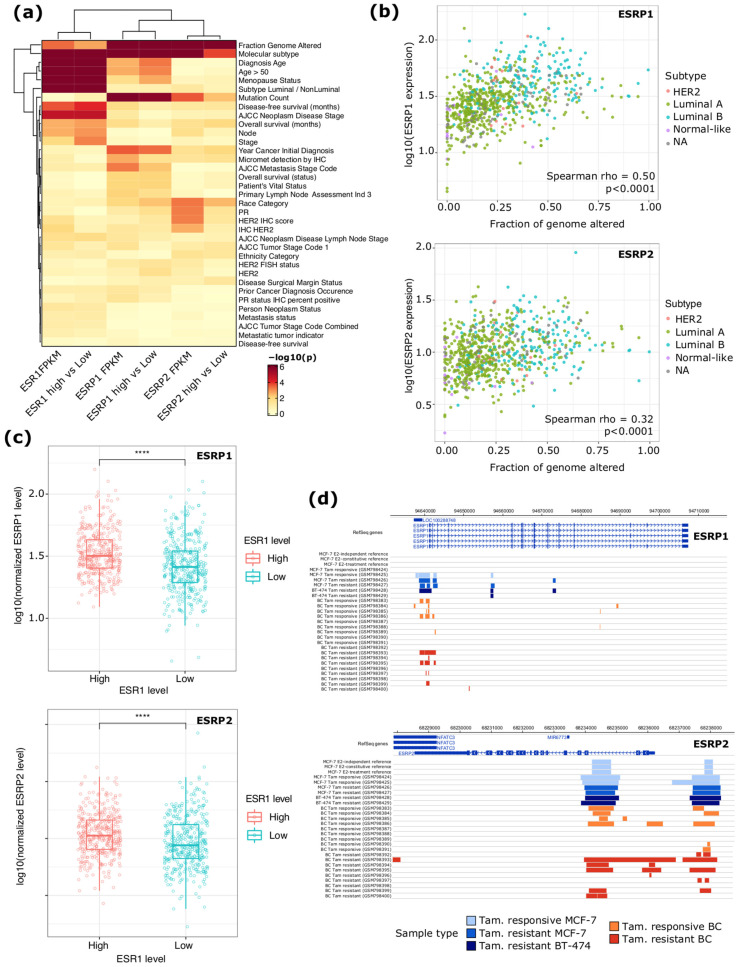
(**a**) Heat map reporting the significance of the association between TCGA ERα+ BC patient clinical data and ESR1, *ESRP1*, and *ESRP2* mRNA levels. The analysis was performed by considering expression levels (1st, 3rd, and 5th columns) or by stratifying in “high” and “low” based on the median gene expression (2nd, 4th, 6th columns). PR—progesterone receptor; IHC—immuno-histochemistry; HER2—human growth factor receptor 2. (**b**) Scatter plots reporting, on the y-axis, the expression level of *ESRP1* (top) and *ESRP2* (bottom) in relation to the degree of genome alteration of TCGA BCs (x-axis). Samples are color coded based on their BC molecular subtype. (**c**) Box plots reporting the *ESRP1* (top) and *ESRP2* (bottom) expression levels in ER+ TCGA BCs partitioned based on ESR1 mRNA levels (e.g., high vs. low). *p*-value by Wilcoxon Rank-Sum test. ****, *p* < 0.0001. (**d**) Genome browser view of *ESRP1* (top) and *ESRP2* (bottom) loci with the indication of the coordinates of ERα ChIP-Seq binding sites in BC cell lines and primary tissues. Data of Tamoxifen (Tam)-sensitive cell lines and tissues are color coded in light blue and orange, respectively. Data from Tam-resistant cell lines/tissues are color coded in blue (MCF-7), dark blue (BT-474), and red (primary BC tissues).

**Figure 2 ijms-23-07835-f002:**
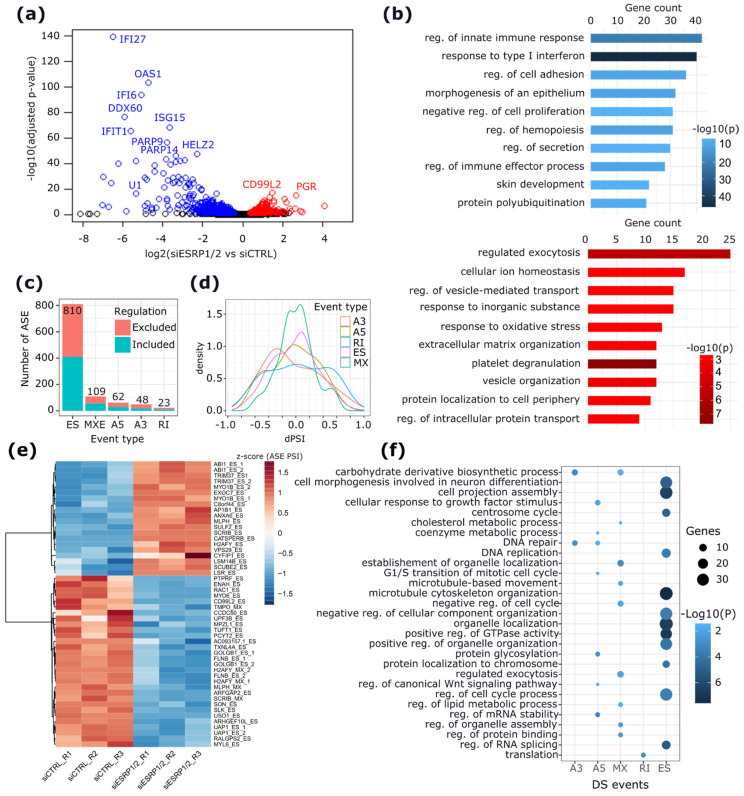
The effects of ESRP1/2 silencing on the transcriptional activity in MCF-7 BC cells. (**a**) Volcano plot reporting the log2FC and significance of downregulated (in blue) and upregulated (in red) DE genes. Labels are provided for the top significant DE genes. (**b**) Bar plots showing the gene counts and the significance of gene ontology (GO) terms enriched for downregulated (blue) and upregulated (red) DE genes. (**c**) Bar plot showing the number of significantly changing ASEs compared to control cells. ES—exon skipping; MX—mutually exclusive exons; A5—alternative 5′ splice sites; A3—alternative 3′ splice sites; RI—retained introns. (**d**) Density plots showing the distribution of delta-PSI (dPSI) of each event type. (**e**) Heatmap of PSI values of the top 50 significant differential ASEs regulated by ESRP1/2 knock-down. The labels provided are the symbols of genes harboring significant ASEs. (**f**) Dot plots showing the GO terms enriched for genes harboring significant ASEs. The size of the dots is proportional to the gene count and the color intensity is proportional to the enrichment significance.

**Figure 3 ijms-23-07835-f003:**
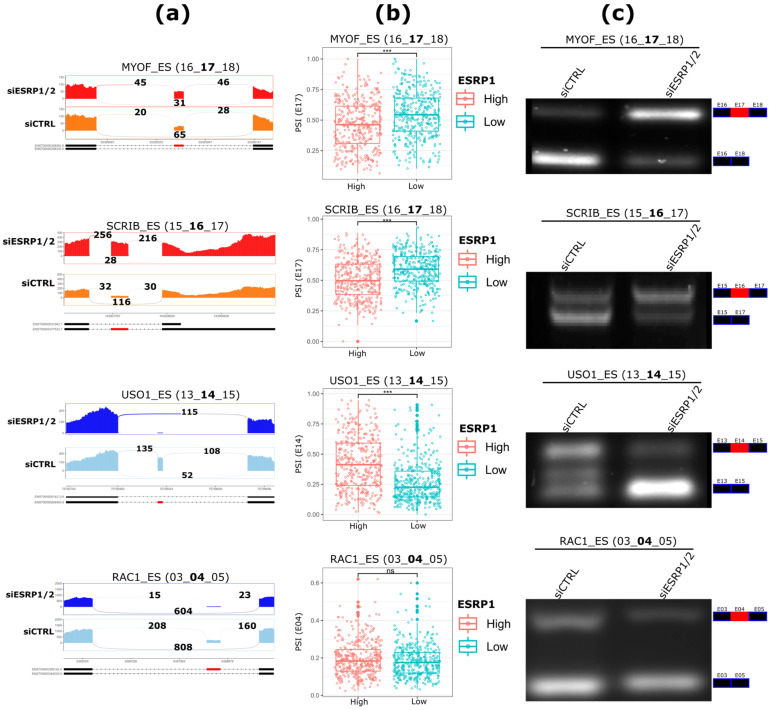
ASEs validation by RT-PCR analysis. (**a**), Sashimiplots reporting the read coverage over the regulated exons and their flanking junctions, as identified by rMATS. The total number of normalized read counts supporting each exon–exon junction is reported. (**b**) The inclusion levels (PSIs) of the validated exons in ER+ BCs partitioned based on ESRP1 median mRNA levels into high- and low-expressing samples. (**c**), RT-PCR products were analyzed using agarose gel electrophoresis. The upper and lower bands represent the inclusion and exclusion forms of the events, respectively. Representative pictures out of three biological replicates. The complete gel image of (**c**) is reported as Appendix A. *** Wilcoxon test *p*-value < 0.001; ns: non significant.

**Figure 4 ijms-23-07835-f004:**
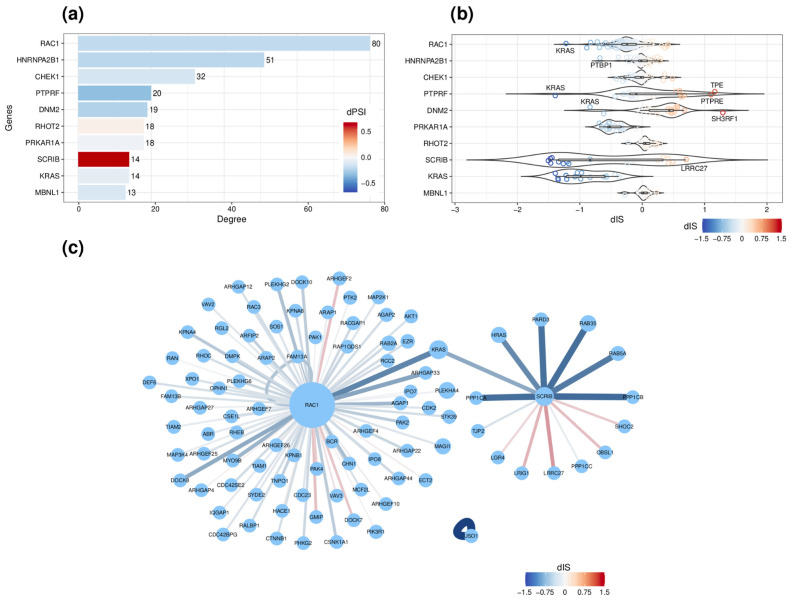
Summary of the protein interaction network and their features derived from the silencing of ESRP1/2. (**a**) Top 10 hub nodes ordered by degree of subnetwork, consisting of genes with significant ASE in our analysis and whose exon inclusion/exclusion levels correlated with ESRP1/2 levels in ER+ BC. The color code of the bars is based on the dPSI of the most significant ES event characterizing each gene. (**b**) Violin plot showing the distribution of dIS of each hub shown in (**a**). (**c**) A highlight of the most changing dIS values in subnetwork, focusing on the PCR-validated targets RAC1, SCRIB, and USO1 and their interactions. The color code of the interactions is based on the dIS value, and width is proportional to the absolute value of dIS.

**Figure 5 ijms-23-07835-f005:**
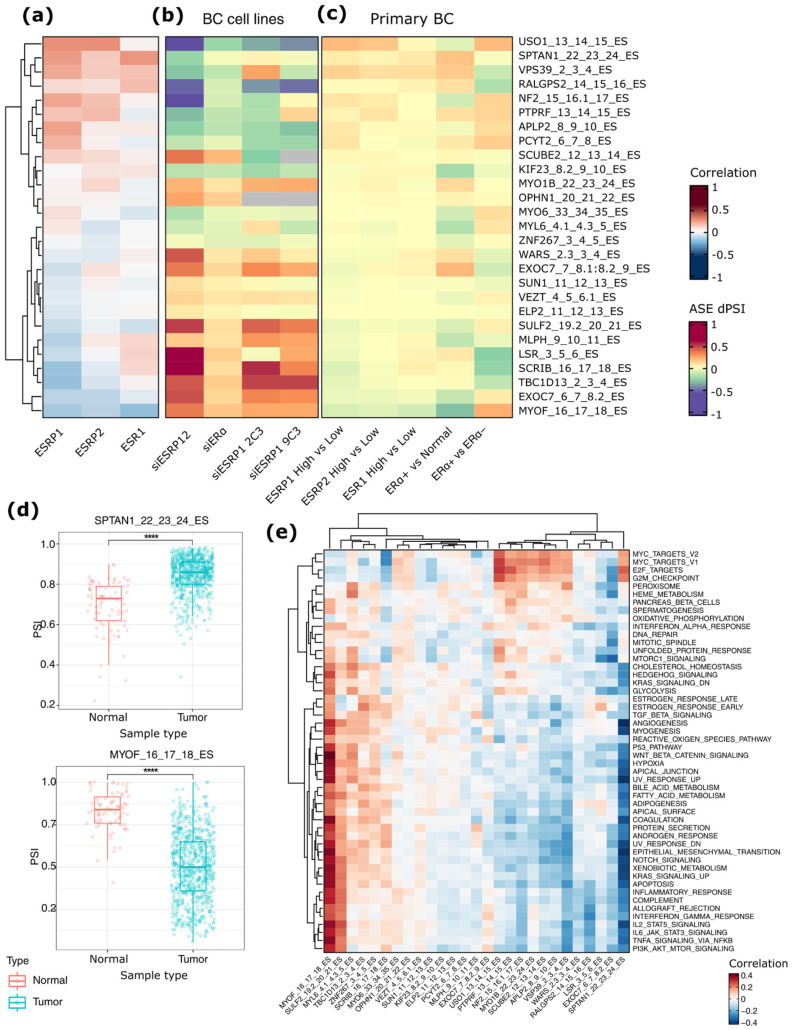
Analysis of the 27 ASEs overlapping between the two experiments in tumor data. (**a**) heat map plot showing the correlation between ESRP1, ESRP2, and ESR1 genes expression and the expression of ASEs exons in tumor samples, whose inclusion changes (dPSI values) upon ESR1 or ESRP1/2 silencing in wild type (MCF-7) or resistant (LCC2, LCC9) BC cell lines are reported in (**b**). (**c**) Heat map plot showing the inclusion levels of the 27 ASEs in primary BC samples classified according to the expression levels of ESRP1, ESRP2, and ESR1 genes. (**d**) Box plots representing the inclusion levels (percent spliced-in index—PSI) of exon 22 of SPATIN1 and exon 16 of MYOF genes in tumor versus normal samples (****, *p* < 0.0001). (**e**) Heat map showing the correlation scores between ASEs and the enriched molecular pathways obtained using PEGASAS [29].

## Data Availability

The raw and processed data analyzed in this study are provided in GEO with the identifier GSE32222.

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
