# Peer review of "A Regulatory Axis between Epithelial Splicing Regulatory Proteins and Estrogen Receptor α Modulates the Alternative Transcriptome of Luminal Breast Cancer"

_ijms, 2022, doi:10.3390/ijms23147835_

Round 1

Reviewer 1 Report

In this manuscript, the authors have explored the functional roles of two epithelial splicing regulators ESRP1 and 2 in ERα+ BCs. The study gives an insight into the ERα-ESRP1/2 axis in the onset and progression of BC by controlling the splicing patterns of related genes. The manuscript is well conceived and presented. However, I have a few points to be addressed.

In figure 1(a), what does ESR1 high vs low mean? How was the stratification achieved?

Figure 1(d) - what is the range of the binding? The Y axis?

Figure 3(c) will benefit from a ladder in each gel image.

Do you have information on the copy numbers of ESRP1 and 2 in the samples you performed the analysis on? 

You have performed combined silencing of ESRP1/2. Does silencing either of them show only modest effect?

Does the AS changes induced upon ESRP1/2 combined silencing revert back rescuing of the proteins?

Were you able to reciprocate the results in other ER-positive cell lines?

There are few typos/spell check required.

Author Response

Reviewer: In this manuscript, the authors have explored the functional roles of two epithelial splicing regulators ESRP1 and 2 in ERα+ BCs. The study gives an insight into the ERα-ESRP1/2 axis in the onset and progression of BC by controlling the splicing patterns of related genes. The manuscript is well conceived and presented. However, I have a few points to be addressed.

Author: We thank the Reviewer for the suggestions that we completely address. A detailed point-to-point response to each comment is reported in the following.

Reviewer: In figure 1(a), what does ESR1 high vs low mean? How was the stratification achieved?

Author: We thank the Reviewer for his question. In figure 1(a), all patients included in the analysis were labeled as “ER-alpha positive” based on the immuno-histological staining in the TCGA specifications (Supplementary Table 1a). Then, these patients were partitioned into (ESR1-positive high) versus (ESR1-positive low) subgroups, referring to the median RNA expression levels of ESR1 gene. Samples with an RNA expression value less than this median value were called “ESR1-positive low”, while samples with a higher expression value were labeled as “ESR1-positive high”. A detail of this stratification was added in the figure caption.

Reviewer: Figure 1(d) - what is the range of the binding? The Y axis?

Author: The figure 1d reports the coordinates of significant peaks obtained from the ChIP-Seq experiments. The ChIP-Seq coverage was not reported. In the revised version of the figure caption, we added a specification about the reported result.

Reviewer: Figure 3(c) will benefit from a ladder in each gel image.

Author: We agree with the Reviewer that this information is lacking in the original version of the figure. To provide this information we added the image of the complete gel, with information about the size of amplified fragments, as the novel Supplementary Figure 5.

Reviewer: Do you have information on the copy numbers of ESRP1 and 2 in the samples you performed the analysis on?

Author: To explore the copy number status of ESRP1 and ESRP2 genes in MCF-7, we explored the CNV data from the Cancer Cell Lines Encyclopedia [1]. This data indicates ESRP1 and ESRP2 to have, respectively, a CNV status (GISTIC score) equal to 2 and 0 in MCF-7 cells (e.g., 2, strongly amplified; 1, amplified; 0, normal; -1, depleted; -2, deeply depleted). Thus, ESRP1 is strongly amplified in the MCF-7 analyzed by this CCLE dataset. In addition, among the 51 BC cell lines with copy number alteration data in this database, 32 were associated with an ESRP1 amplification (62.54%) while only 14 with an ESRP2 copy gain (27.45%). In the revised version of the main text, we added this information in the discussion section with details in the novel Supplementary Table 9. Conversely, for the breast cancer samples from the TCGA cohort, we reported the CNV status in Supplementary Table 1a (columns J/K). The observed over-amplification of ESRP1 gene locus in BC was discussed in detail in the discussion section, showing agreement with previous studies [2].

Reviewer: You have performed combined silencing of ESRP1/2. Does silencing either of them show only modest effect?

Author: We thank the Reviewer for his interesting question. To get an optimal effect of the silencing, for the present publication we decided to perform a combined silencing of both factors and did not try to silence either one of them. However, previous studies [3] have shown that indeed silencing either of the two had only limited/modest effects on gene expression and alternative splicing in general. Furthermore, a study by Ishii and colleagues [4] has illustrated two different mechanisms by which ESRP1 and ESRP2 regulate cell motility of cancer cells. Silencing ESRP1 affected the dynamics of the actin cytoskeleton by depressing the expression of Rac1b isoform, whereas the knockdown of ESRP2 attenuates the cell-to-cell contact by increasing the expression of epithelial-to-mesenchymal transition associated transcription factors, like δEF1/SIP1. These data support the use of a double knockdown experiment to achieve the best silencing effect.

Reviewer: Does the AS changes induced upon ESRP1/2 combined silencing revert back rescuing of the proteins?

Author: We thank the Reviewer for his interesting question. We did not perform a rescue experiment of ESRP1/2 in this study to exclude the off-target effects of the combined silencing of these factors. However, in our analysis a consensus list of AS events previously reported as targets of ESRP1/2 factors was controlled and validated. These results were provided in Supplementary Figure 3. In addition, we confirmed in primary tumors the significant correlations between the levels of expression of these factors and the levels of inclusion of 122 AS events, representing 37% of the total events detected in these tumors. This data is reported in Supplementary Table 5c.

Reviewer: Were you able to reciprocate the results in other ER-positive cell lines?

Author: At our laboratory, we did not explore the effects of silencing ESRP1/2 on the alternative transcriptome of other ER-positive cell lines. However, in the present version of the manuscript we compared our list of significant AS changes with those observed upon ESRP1 silencing in the ER-positive, endocrine-resistant, 2C3 and 9C3 BC cell lines (Supplementary Table 8e). Furthermore, 171 and 109 AS events that occurred in our dataset were confronted with those occurring upon ESRP1/2 silencing in two independent RNA seq studies using different cellular models (Supplementary Figure 11), showing a very strong agreement.

Reviewer: There are few typos/spell checks required.

Author: We carefully revised the text and corrected any possible typos.

References

  1. Barretina, J.; Caponigro, G.; Stransky, N.; Venkatesan, K.; Margolin, A.A.; Kim, S.; Wilson, C.J.; Lehár, J.; Kryukov, G.V.; Sonkin, D.; et al. The Cancer Cell Line Encyclopedia Enables Predictive Modelling of Anticancer Drug Sensitivity. Nature 2012, 483, 603–607, doi:10.1038/nature11003.
  2. Sebestyén, E.; Singh, B.; Miñana, B.; Pagès, A.; Mateo, F.; Pujana, M.A.; Valcárcel, J.; Eyras, E. Large-Scale Analysis of Genome and Transcriptome Alterations in Multiple Tumors Unveils Novel Cancer-Relevant Splicing Networks. Genome Res. 2016, 26, 732–744, doi:10.1101/gr.199935.115.
  3. Shapiro, I.M.; Cheng, A.W.; Flytzanis, N.C.; Balsamo, M.; Condeelis, J.S.; Oktay, M.H.; Burge, C.B.; Gertler, F.B. An EMT-Driven Alternative Splicing Program Occurs in Human Breast Cancer and Modulates Cellular Phenotype. PLoS Genet. 2011, 7, e1002218, doi:10.1371/journal.pgen.1002218.
  4. Ishii, H.; Saitoh, M.; Sakamoto, K.; Kondo, T.; Katoh, R.; Tanaka, S.; Motizuki, M.; Masuyama, K.; Miyazawa, K. Epithelial Splicing Regulatory Proteins 1 (ESRP1) and 2 (ESRP2) Suppress Cancer Cell Motility via Different Mechanisms. J. Biol. Chem. 2014, 289, 27386–27399, doi:10.1074/jbc.M114.589432.

Reviewer 2 Report

In this article, the authors focused on two splicing regulators, ESRP1 and ESRP2, and their involvement in the control of the transcriptome of ERα+ cells, as well as in breast cancer samples.

After having shown that the expression of the two genes is dependent on the ERα receptor, the authors analyzed the effect of a double knock-down of the two proteins ESRP1 and ESRP2 on the transcriptome of MCF-7 cells, as well as the alternative splicing. Changes in ASEs are clearly established and the consequences of ASEs are studied as a network of protein-protein interactions. All the results clearly show a strong involvement of the two proteins ESRP1 and 2, in association with the ERα receptor, on the aggressiveness of ERα positive breast cancer.

This article is very well constructed, with a relevant scientific approach and clear and precise results. Figure 1, however, needs to be enlarged.

Author Response

Reviewer: In this article, the authors focused on two splicing regulators, ESRP1 and ESRP2, and their involvement in the control of the transcriptome of ERα+ cells, as well as in breast cancer samples.

After having shown that the expression of the two genes is dependent on the ERα receptor, the authors analyzed the effect of a double knock-down of the two proteins ESRP1 and ESRP2 on the transcriptome of MCF-7 cells, as well as the alternative splicing. Changes in ASEs are clearly established and the consequences of ASEs are studied as a network of protein-protein interactions. All the results clearly show a strong involvement of the two proteins ESRP1 and 2, in association with the ERα receptor, on the aggressiveness of ERα positive breast cancer.

This article is very well constructed, with a relevant scientific approach and clear and precise results. Figure 1, however, needs to be enlarged.

Author: We thank the Reviewer for his insightful comment. In the revised version of the manuscript, we enlarged Figure 1.

Reviewer 3 Report

A manuscript “Regulatory axis between Epithelial Splicing Regulatory Proteins and Estrogen Receptor alpha modulates the alternative transcriptome of luminal breast cancer” by Jamal Elhasnaoui, Giulio Ferrero, Valentina Miano, Lorenzo Franchitti, Isabella Tarulli, Lucia Tarrero Coscujuela, Santina Cutrupi, and Michele De Bortoli is an excellent work. The silencing of ESRP1/2 in MCF-7 cells and RNA-Seq showed the dysregulation of more than 700 genes and alternative splicing of genes involved in cell signaling, metabolism, cell growth, and EMT. The manuscript is well-written. All analyses are well conducted. Results may be of interest to both clinicians and basic scientists.

Minor remark:

Lines 519-522: The analysis of the gene expression levels in relation to continuous clinical data was performed using a Spearman correlation analysis (implemented in the cor_test R function) while categorical variables were analyzed using a Kruskal-Wallis test (implemented in the kruskal.test R function).

Please reword this sentence. These were the comparisons of continuous variables between groups that were analyzed using a Kruskal-Wallis test

Author Response

Reviewer: A manuscript “Regulatory axis between Epithelial Splicing Regulatory Proteins and Estrogen Receptor alpha modulates the alternative transcriptome of luminal breast cancer” by Jamal Elhasnaoui, Giulio Ferrero, Valentina Miano, Lorenzo Franchitti, Isabella Tarulli, Lucia Tarrero Coscujuela, Santina Cutrupi, and Michele De Bortoli is an excellent work. The silencing of ESRP1/2 in MCF-7 cells and RNA-Seq showed the dysregulation of more than 700 genes and alternative splicing of genes involved in cell signaling, metabolism, cell growth, and EMT. The manuscript is well-written. All analyses are well conducted. Results may be of interest to both clinicians and basic scientists.

Minor remark:

Lines 519-522: The analysis of the gene expression levels in relation to continuous clinical data was performed using a Spearman correlation analysis (implemented in the cor_test R function) while categorical variables were analyzed using a Kruskal-Wallis test (implemented in the kruskal.test R function).

Please reword this sentence. These were the comparisons of continuous variables between groups that were analyzed using a Kruskal-Wallis test

Author: We thank the Reviewer for his insightful comments. Specifically, the correlation analysis was performed between continuous variables and the target gene levels. The Kruskall-Wallis’s test was applied to test the difference in target gene expression (ESR1, ESRP1, or ESRP2) in patients stratified based on specific covariates (e.g., the breast cancer molecular subtypes). In the revised version of the manuscript, we rewrote the sentence to make it clearer.